# Deformation and Phase Transformation of Disordered α Phase in the (α + γ) Two-Phase Region of a High-Nb TiAl Alloy

**DOI:** 10.3390/ma14174817

**Published:** 2021-08-25

**Authors:** Haitao Zhou, Fantao Kong, Yanbo Wang, Xiangwu Hou, Ning Cui, Jingli Sun

**Affiliations:** 1Research & Development Center, Shanghai Spaceflight Precision Machinery Institute, Shanghai 201600, China; wybyanyu@163.com (Y.W.); hou_xiangwu@163.com (X.H.); 2School of Materials Science and Engineering, Harbin Institute of Technology, Harbin 150001, China; kft@hit.edu.cn; 3School of Mechanical and Automotive Engineering, Qingdao University of Technology, Qingdao 266520, China; cuining@qut.edu.cn

**Keywords:** TiAl alloy, disordered α phase, deformation, phase transformation, continuous dynamic recrystallization, rolling

## Abstract

In this paper, the deformation and phase transformation of disordered α phase in the (α + γ) two-phase region in as-forged Ti-44Al-8Nb-(W, B, Y) alloy were investigated by hot-compression and hot-packed rolling. The detailed microstructural evolution demonstrated that the deformed microstructure was significantly affected by the deformation conditions, and the microstructure differences were mainly due to the use of a lower temperature and strain rate. Finer α grains were formed by the continuous dynamic recrystallization of α lamellae and α grains distributed around lamellar colonies. Moreover, the grooved γ grains formed by the phase transformation from α lamellae during hot rolling cooperated with and decomposed α lamellae. A microstructure evolution model was built for the TiAl alloy at 1250 °C during hot rolling.

## 1. Introduction

TiAl alloys are of great interest in high-temperature applications due to their low density, high specific strength, oxidation resistance, and good structural stability at high temperatures [1,2]. In recent years, research has shifted toward high-Nb TiAl alloys due to their higher service temperatures, which makes them attractive for replacing some Ni-based superalloys in applications such as gas turbine engines [3,4,5]. Due to their attractive properties, TiAl alloy sheets can be used in aerospace applications such as inlet flaps, nozzle tiles for gas turbine engines, back structures, and heat shields of jet aircraft [6,7,8]; however, their poor high-temperature workability remains a major bottleneck for the successful application of high-Nb TiAl alloys [9,10,11].

Some studies have shown that thermal processing is an attractive and high-efficiency way to improve the ductility and workability of alloys via microstructural refinement and homogenization [12]. In terms of the manufacturing of TiAl alloy sheets, ingot metallurgical processes were developed and applied to a variety of alloy systems [9,12,13,14]. Cast ingots with fully lamellar or nearly lamellar colonies must undergo a series of thermal deformations to break down coarse lamellar colonies. To date, the deformation behavior and mechanism of lamellar colonies have been studied in detail. For example, the interruption, rotation, bending, kinking, and breaking down of lamellar colonies are responsible for the microstructure evolution of Ti-45Al-8.5Nb-(W, B, Y) alloy [13]. Dynamic recrystallization, superplastic deformation, and mechanical twinning play critical roles in the hot-rolling of powder metallurgy Ti-45Al-7Nb-0.3W alloy [15]. The bending and fracturing of lamellar structures and dynamic recrystallization are the main mechanisms responsible for flow softening during the later stages of hot-pack rolling of Ti-47Al-2Cr-0.2Mo alloy [16]. The recovery and recrystallization of γ phases, as well as the spheroidization of α phases, play important roles during the refining of the microstructure of Ti-44Al-8Nb-(W, B, Y) alloy [17]. Dynamic recrystallization of γ grains is the main softening mechanism during the hot forging of the Ti-43Al-2Cr-2Mn-0.2Y alloy [18]. The coarsening of γ lamellae was caused by the lath decomposition of the Ti-44Al-6V-3Nb-0.3Y alloy [14]. Almost all literature has paid more attention to the transformation of γ phases or has studied lamellar colonies but has neglected the role of the disordered α phase in deformation.

Before hot deformation, TiAl alloys are often annealed in the (α + γ) phase region for hours [8,9,13], which transforms the γ phase into disordered α phase; thus, TiAl alloys will often contain a large amount of disordered α phase at the deformation temperature, which plays an important role in the deformation process. So far, the deformation behavior of the disordered α phase has rarely been documented. In this paper, the deformation behavior of the disordered α phase in a high-Nb TiAl alloy was investigated by hot-compression and hot-packed rolling. The aim of this work was to clarify the deformation and phase transformations of α lamellae and α grains in the (α + γ) two-phase region.

## 2. Experimental

The nominal Ti-44Al-8Nb-(W, B, Y) alloy casting ingot was prepared by induction skull melting (ISM). The true chemical composition of the high-Nb TiAl alloy was Ti-43.9Al-8.1Nb-0.18W-0.2B-0.3Y (at %). The alloy presents a uniform near lamellar microstructure and consists of lamellar colonies, γ and α_2_ grains, a small amount of borides with a curvy morphology and Al_2_Y in bulk particles [19]. After homogenization heat treatment at 900 °C for 48 h, hot isostatic pressuring (HIP) was conducted for 4 h at 1250 °C under a pressure of 80 MPa in an Ar atmosphere. Then, the forged billet with dimensions of Φ 550 × 50 mm^3^ was obtained by quasi-isothermal forging of the ingot at 1250 °C with a reduction of 75%. The microstructure of the as-forged alloy had a nearly lamellar microstructure composed of many lamellar colonies with a mean colony size of 120 μm and a small amount of 10 μm fine equiaxed γ grains along its boundaries [9].

Cylindrical specimens for isothermal compression with dimensions of Φ8 × 12 mm^3^ and plate specimens with dimensions of 40 × 40 × 12 mm^3^ for hot-packed rolling were cut from the as-forged Ti-44Al-8Nb-(W, B, Y) alloy pancake along the forging direction. The plate specimens were cleaned by grinding and then encapsulated in stainless steel to fabricate the packed specimens. Both the cylindrical and packed specimens were heat-treated at 1250 °C for 2 h before hot deformation and then compressed to 58% of their original height or packed by three deformations with a reduction of 25% per pass. The nominal strain rate was 0.1 s^−1^ for isothermal compression and 0.5 s^−1^ for hot-packed rolling, respectively. The reheating time between two deformations was 15 s and 15 min for isothermal compression and hot-packed rolling, respectively. The heat-treated and deformed samples were immediately quenched in ice water to preserve the deformed microstructure and then sectioned through their longitudinal axis for microstructure analysis.

Microstructures were analyzed via electron backscatter diffraction (EBSD) and transmission electron microscopy (TEM). EBSD measurements were performed using a Hikari camera (EDAX) mounted in an FEI Quanta 200FEG scanning electron microscope at scanning intervals of 0.2 μm. All EBSD data were analyzed using IOM 5.31 software. The EBSD specimens were electrolytically polished using a solution of 6% perchloric acid + 34% butanol + 60% methanol at −20 °C and 25 V. The foils for TEM observations were mechanically polished to 60–80 μm and then polished again with a twin-jet electrolytic apparatus using the same polishing solution as was used to prepare EBSD specimens. In the following sections, α_2_ is expressed as the ordered α phase, and α represents the disordered α phase at high temperatures. All orientation maps used the inverse pole figure coloring scheme relative to the *x*-direction.

## 3. Results and Discussion

The EBSD analysis of the as-forged microstructure is shown in Figure 1. It can be seen from Figure 1a that the microstructure of the as-forged alloy was mainly composed of coarse lamellar colonies (average size of 120 μm) and equiaxed γ grains distributed around the lamellar colonies. The details regarding the microstructure for as-forged alloys have been given previously in ref. [12]. These equiaxed γ grains with high-angle grain boundaries (blue line) around lamellar colonies are recrystallized grains produced during forging. The partition fraction of the γ phase in the as-forged alloy was about 82 vol.%, and that of the α_2_ phase was only about 18 vol.%. The majority of α_2_ lamellae and γ lamellae in the lamellar colony remained as lamellae. As can be seen from Figure 1b, the orientation of the α_2_ lamellae inside the lamellar colony was substantially unchanged. By observing the pole figure of α_2_ lamella, γ-1 lamella (marked with a red line), and γ-2 lamella phase in Figure 1b,c, it can be found that the (111)γ-1 planes were substantially parallel to the (0001)α_2_ plane, and the <1$\bar{1}$0]γ-1 direction was substantially parallel to the <11$\bar{2}$0>α_2_ direction, following a Blackburn orientation relationship. While the γ-2 lamella does not follow such a relationship with the α_2_ lamella. Moreover, the γ-2 lamella (the black region in Figure 1a) contains more low-angle grain boundaries (red line) than the γ-2 lamella (the white region in Figure 1a).

The microstructure of the as-quenched alloy determined by EBSD analysis after annealing at 1250 °C for 2 h is shown in Figure 2. The microstructure was mainly composed of lamellar colonies and equiaxed α grains and γ grains around these lamellar colonies. The detailed SEM analysis of the microstructure of quenched Ti-44Al-8Nb-(W, B, Y) alloy after annealing is given in ref. [9]. The phase transition of γ phase into α phase and the transformation of ordered α_2_ phase into disordered α phase occurred during annealing in the (α + γ) two-phase region, which increased the α phase content. Figure 2a shows that after annealing, the partition fraction of disordered α phase with equiaxed and lamellae morphology in the as-heat-treated alloy was more than 70 vol.%. So, the sample contained more than 70 vol.% disordered α phase before hot-packed rolling, which was softer than the ordered α_2_ and γ phases and more easily deformed, as confirmed by thermal deformation [9,19].

From the phase diagram of Ti-Al alloys [20] and Ti-Al-Nb alloys [21], it can be seen that the (α + γ) two-phase region exists above the *T_e_* temperature, and the α phase in this region is disordered, which has good deformability. Upon increasing the deformation temperature in the (α + γ) two-phase region, the deformation microstructure contains a higher content of disordered α phase and a higher percentage of equiaxed α grains and α lamellae. During deformation, in addition to the refinement of these γ grains, disordered α grains and α lamellae are also transformed into even finer α grains. Moreover, when there are more α grains and α lamellae in the original microstructure, there are also more (α_2_ + γ) lamellar colonies in the deformed microstructure. When the deformation temperature is close to *T*_α_, the microstructure contains many disordered equiaxed α grains, which transform into fine equiaxed α grains during deformation, resulting in a fully-lamellar microstructure; therefore, the deformation microstructure strongly depends on the contents of α and γ phases, i.e., on the deformation temperature. Refined and uniform duplex, nearly lamellar, and fully lamellar microstructures can be directly obtained after hot rolling [22].

Al is an important α phase stabilizing element, and reducing the Al content can improve the deformability of TiAl alloys by introducing a large amount of disordered α phase under high-temperature deformation. For the TiAl alloy in this paper, the Al content was about 44%. There is a higher disordered α phase content for the TiAl alloy at 1250 °C than other high-Nb TiAl alloys with higher Al contents such as Ti-45Al-8.5Nb-(W, B, Y), Ti-46Al-9Nb, and Ti-45Al-7Nb-0.3W alloys. Therefore, its deformability at 1250 °C is superior to other high-Nb TiAl alloys. The current alloy still has a higher peak flow stress due to its high Nb content and nearly lamellar structure, confirmed by hot compression experiments [17]. The higher-content α grains and α lamellae at (α + γ) two-phase region play an important role in the transformation of the lamellar colonies.

Figure 2b,c show orientation maps of the α and γ phases, respectively. It can be found that some α lamellae and γ lamellae in the lamellar colonies did not follow a strict orientation relationship, and there is some deviation from the standard orientation between γ lamellae and α lamellae—especially the γ lamellae. The orientation difference between α_2_-1 lamella and α_2_-2 lamella in Figure 2b is 2.9°, and the orientation difference between γ-3 lamella and γ-4 lamella in Figure 2c is as high as 13°. This indicates that the orientation relationship of α lamellae and γ lamellae was not changed due to a phase transformation during isothermal holding at 1250 °C, and it still maintained the orientation relationship formed during hot forging. The lamellae that do not follow a Blackburn orientation relationship or that contain many low-angle grain boundaries more easily undergo recovery and recrystallization. 

Figure 3 shows the microstructure of the as-quenched alloy after isothermal compression. After isothermal compression, lamellar colonies underwent significant and thorough decomposition and were completely transformed into slightly elongated γ and α grains perpendicular to the compression direction. The orientation of each α grain marked with a 3D grain orientation in Figure 3b shows that these α grains had orientations that were significantly different from other α grains. This indicates that dynamic recrystallization was more likely to occur for the disordered α phase under these deformation conditions. Moreover, the total partition fraction of the α phase in the as-deformed alloy was not significantly changed compared with that of the as-heat-treated alloy and remained at about 73%, as shown in Figure 3a,c,d show the orientation and pole figures of α grains and their surrounding γ grains. The finer α grains and surrounding γ grains do not follow a Blackburn orientation relationship, implying that no phase transformation from α phase to γ phase occurred during isothermal compression.

The microstructure of the water-quenched alloy after hot-packed rolling is shown in Figure 4. It contains many recrystallized grains and remnant lamellar colonies (marked with a black line). The orientation difference between α lamellae in the remnant lamellar colony was obviously increased compared with that in the as-heated-quenched alloy. The misorientation between αL1 lamella and αL2 lamella was as high as 16.5°, as shown in Figure 4b.

Figure 5 shows the misorientation measured along the L1 line parallel to the αL1 lamella and along the H1 line perpendicular to the αL1 lamella in Figure 4b. The accumulative (point-to-origin) misorientations in the L1 direction gradually increased to 7.4°, while the largest accumulative angle along the H1 direction did not exceed 1°. This indicates that the rotation degree along the length of the αL1 lamella was greater than that along the width. These α lamellae with a higher misorientation will absorb deformation energy during the next rolling pass, further promoting the rotation and increasing the misorientation along the length of the α lamellae. Some α sub-grains will form along the length of α lamellae during the rotation of α lamellae, and then these α sub-grains can rotate, which increases the orientation difference. Eventually, some new α grains will form along the length of the α lamellae, such as the new α grain shown by the arrow in Figure 4b. The above process is the evolution of α lamellae into α grains, i.e., continuous dynamic recrystallization (CDRX) of α phase.

In addition to parts of α grains formed by the CDRX of α lamellae, more α grains were formed by the CDRX of α grains distributed around lamellar colonies during deformation. The CDRX of parent α grains occurred around lamellar colonies, and the original coarse α grains transformed into finer α grains after three passes of hot-packed rolling, as shown in Figure 6a, such as region 1 in Figure 4a. These newly-formed α grains have specific orientations with each other, e.g., the misorientations among α_2_-3 grain and α_2_-4, α_2_-5, α_2_-6, and α_2_-7 grains were 14.5°, 16.9°, 5.8°, and 28°.

It also can be seen in Figure 4a that after hot-packed rolling, the partition fraction of the α phase decreased to about 50% with a reduction of about 20%. The fabrication of TiAl alloy sheets was not an isothermal process but was accompanied by a temperature drop. The temperature drop during rolling is the main reason for the decrease in the α phase content. Figure 6b shows the pole figures of α_2_-3 grain and γ-5 and γ-6 grains around α_2_-3 grain in Figure 6a. Comparing (111)γ with (0001)α_2_ pole figures, and <1$\bar{1}$0]γ with <11$\bar{2}$0>α_2_ pole figures between γ-5 and α_2_-3, the (111)γ planes were substantially parallel to the (0001)α_2_ plane, and the <1$\bar{1}$0]γ direction was substantially parallel to the <11$\bar{2}$0>α_2_ direction, following a perfect orientation relationship of (111)γ//(0001) α_2_ and <1$\bar{1}$0]γ//<11$\bar{2}$0>α_2_. This indicates that the phase transformation from α grain to γ grain was promoted by the temperature drop during hot-packed rolling, which decreased the α phase content. Moreover, the curvature of the γ-5 grain in Figure 6a proves the occurrence of α phase transformation. Yet, there are some differences in the perfect orientation relationship between γ-6 and α_2_-3 grains, which are mainly attributed to grain rotation during hot-packed rolling.

In addition to the transformation of α phase to γ phase in α grains, these phase transformations can also occur in α lamellae. Furthermore, the newly-formed γ grains and α lamella also follow a Blackburn orientation relationship, as shown in Figure 6c. The precipitation and growth of γ grains in α lamellae were found in remnant lamellar colonies of Ti-45Al-10Nb alloys [23]. Eutectoid transformation can also occur in recrystallized α grains, as confirmed by the orientation relationship in Figure 6d. It can be concluded from the above analysis that complex deformation and phase transformations occur in the disordered α phase during hot-packed rolling. Disordered α grains and α lamellae will first deform into recrystallized α grains, and then now γ grains or γ lamellae will form in these recrystallized α grains. The discontinuous dynamic recrystallization (DDRX) of the γ phase occurs at tri-boundaries, phase boundaries, and twin boundaries, as confirmed by previous research [12].

The TEM analysis of the water-quenched as-rolled Ti-44Al-8Nb-(W, B, Y) alloy is shown in Figure 7. Analysis of the lamellar structure in Figure 7a shows that the lamellar structure mainly consisted of α lamellae with an unbalanced interface. During high-temperature plastic deformation, the α phase was softer than the γ phase and can coordinate the deformation of the γ phase, although it still has a high stacking fault energy. To coordinate the deformation of the “hard” γ phase, bending, kinking, or continuous dynamic recrystallization will occur α lamella. This is demonstrated by the phenomena of the “cut off” of the α lamella by the γ phase, as indicated by the arrow in Figure 7a.

During deformation, γ lamellae are easily broken into new γ grains, as shown in Figure 7b, but it is difficult for the α lamellae to form new α grains in the same way as γ grains (by nucleation and growth). The above analysis shows that there is an increased misorientation in the longitudinal direction of α lamella, which causes rotation along the longitudinal direction to be higher than that in the thickness direction; therefore, some sub-grains were gradually formed along the longitudinal direction of the α lamella upon increasing the rotation. These α sub-grains are pointed out with a white arrow in Figure 7b, and their sub-grain boundaries are mostly perpendicular to the interface of the α/γ lamellae. Then, some new α grains will be formed along the length of the α lamella. Disordered α lamellae cannot decompose into new α grains by themselves and require an external driving force. The schematic diagram of lamellar decomposition is represented in Figure 8. Normally, the decomposition mechanism includes boundary splitting, which results in lamellar fragmentation and termination migration that leads to spheroidization [24].

A groove grain appeared on the boundaries of α/γ lamellae during deformation, which can accelerate the decomposition of disordered α lamella and thus form new α grains, as shown in Figure 7c.

The α lamellae and γ lamellae are indicated by A and B, respectively. There are α lamellae and γ lamellae above the *T*_e_ for Ti-44Al-8Nb-(W, B, Y) alloy; thus, in Figure 8b, *S_AA_* and *S_AB_* are the surface energies of the α/α lamellae interface and α/γ lamellae interface, and 2*θ* is the angle between *S_AA_* and *S_AB_* from Figure 8b. The equilibrium equation between *S_AA_* and *S_AB_* is
*S_AA_* = 2*S_AB_* cos *θ*(1)

The high chemical potential at grooves will promote the migration of atoms away from grooves. So, the curvature radius will continue to increase, resulting in *S_AA_* > 2*S_AB_* cos *θ*. To rebalance the interface energy, the groove in the α lamella was deepened, which increased the angle *θ*. The above process occurs repeatedly until these α lamella fragments. The spheroidization mechanism is similar to that of the termination migration mechanism, which is realized by the diffusion of atoms from a small curvature radius to a large curvature radius to form equiaxed α grains.

The external factor, the newly formed γ grain, accelerates the spheroidization of α lamella, as shown in Figure 7d. As shown in Figure 6c, a grooved γ grain was generated by the phase transition from α lamella during hot rolling. Since disordered α lamellae are soft above the *T*_e_, γ grains will easily grow into α lamella, which accelerates the decomposition of α lamella. Figure 7e shows the equiaxed α grains formed by the CDRX of α lamella and the assisted decomposition of γ grains. When the γ grain diffuses through the α lamella, the parent α lamella will be completely separated, and the α lamella is ultimately transformed into non-continuous equiaxed α grains, as shown in Figure 7g.

Figure 9 shows the TEM analysis of the microstructure around lamellar colonies in the water-quenched as-rolled Ti-44Al-8Nb-(W, B, Y) alloy after three rolling passes. On one hand, α lamellae and γ lamellae inside lamellar colonies are converted to α grains and γ grains. On the other hand, α grains and γ grains around lamellar colonies are changed into finer α grains and γ grains. Figure 9a depicts the recrystallized α grains around lamellar colonies and their surrounding γ grains. There is a temperature drop in TiAl alloy sheets fabricated by hot-packed rolling, and CDRX and phase transformations will occur in these equiaxed α grains. The white arrow in Figure 9a indicates the newly-formed γ grains, which were formed by the phase transformation of recrystallized α grains, as confirmed by the diffraction pattern in Figure 9b and the orientation relationship in Figure 6a,b.

In addition, γ lamellae will precipitate from recrystallized α grains, forming finer lamellar colonies as shown in Figure 9c, but the precipitation of γ lamellae is not complete due to water quenching. γ lamellae do not cross the entire α grain, and the γ lamellae are only a few atoms thick, as shown in Figure 9d,e. Figure 9f shows that the γ and α grains appeared around lamellar colonies, and these γ grains include recrystallized γ grains and γ grains transformed from α grains.

The EBSD analysis of the microstructure of water-quenched as-rolled Ti-44Al-8Nb-(W, B, Y) alloy with a total rolling reduction of 58% after annealing at 1250 °C for 15 min is shown in Figure 10. The newly-formed recrystallized α grains recrystallized γ grains, and γ lamellae in Figure 4 transformed into disordered α grains and α lamellae during re-heating, leading to an increase in the content of the disordered α phase to more than 70 vol.%. Therefore, we can conclude that the partition fraction of the disordered α phase at high temperatures was higher than 70 vol.% before each rolling pass. Moreover, the content of disordered α grains increased with the progression of rolling, and the size and volume fraction of the lamellar colonies were significantly reduced after three rolling passes and two reheating processes.

The orientation relationship between α lamella and γ lamella was unchanged, and some still did not display a Blackburn orientation relationship, as seen in the orientation map of α lamella and γ lamella (black outline, Figure 10b. Figure 10c shows the misorientation of α lamella in Figure 10b along the longitudinal direction, which still has a higher accumulative misorientation of 9.2° along the longitudinal direction. This means that α lamellae with a higher misorientation will preferentially transform into new α grains along the longitudinal direction during subsequent rolling. The above analysis shows that CDRX is more likely to occur in disordered α grains than in α lamellae; therefore, the microstructure of the TiAl alloy in Figure 10 has better deformability, and recrystallization will take place in either α grains and γ grains or in α lamellae and γ lamellae. This will lead reduce the size and content of remnant lamellar colonies. 

Based on the above analysis, the microstructure evolution model of the Ti-44Al-8Nb-(W, B, Y) alloy at 1250 °C during hot-packed rolling is represented in Figure 11. First, the transformation of γ phase to α phase and the transformation of ordered α_2_ phase to disordered α phase occurred simultaneously during the heat preservation process in the (α + γ) two-phase region. Figure 11b describes the microstructure of the TiAl alloy before the first rolling pass, which mainly consists of lamellar colonies with coarsened α lamellae and >70 vol.% disordered α phase. These α lamellae, γ lamellae, α grains, and γ grains will simultaneously transform into finer α grains and γ grains during hot rolling. Yet, the degree of conversion of α grains and γ grains is much better than that of α lamellae and γ lamellae. These α grains and γ grains around lamellar colonies undergo thorough recrystallization into fine recrystallized α grains and γ grains. By contrast, the misorientation of α lamellae along the longitudinal direction was increased, and α small amount of α sub-grains or grains were formed by deformation. Moreover, these recrystallized α grains will transform into γ grains and lamellar colonies during rolling due to a temperature drop. In general, the use of a larger rolling deformation per pass or lower rolling rate is favorable for the decomposition of lamellar colonies.

Figure 11d is the microstructure of Figure 11c after holding at 1250 °C for 15 min. This process causes the disordered α phase to increase to more than 70 vol.% before rolling, with the proportion of α grain increased. Moreover, both the size and content of remnant lamellar colonies decreased. The workability of the as-rolled alloy was significantly improved due to increased α grains and decreased lamellar colonies after previous hot rolling; therefore, under the same hot rolling conditions, finer γ grains and newly-formed lamellar colonies formed, as shown in Figure 11e. As deformation progressed, more α lamellae were converted into α grains due to increased misorientation and enhanced assisted decomposition of α lamellae by γ grains. Finally, the original coarse lamellar colonies were converted into fine γ grains and lamellar colonies by repeated rolling deformation and furnace heat preservation. The proportion of residual lamellae and newly-precipitated lamellar colonies in the as-rolled microstructure was related to the rolling process. The proportion of residual lamellar colonies was reduced, and the volume fraction of newly-formed lamellar colonies was increased by reducing the rolling strain rate and increasing the per-pass and total deformation. In the literature, the original coarse nearly-lamellar structure with a mean grain size of 120 μm was converted into a fine duplex microstructure with a mean grain size of 5.3 μm after hot-packing rolling with a large thickness reduction of 85%, which a reduction of 25% per pass [9]. 

## 4. Conclusions

From the data presented concerning the microstructure of as-deformed and as-rolled Ti-44Al-8Nb-(W, B, Y) alloy, the following conclusions were obtained:

The as-deformed microstructure was significantly affected by the deformation conditions. The nearly-lamellar microstructure was completely transformed into slightly elongated γ and α grains after three isothermal compression cycles at 1250 °C/0.1 s^−1^/25%. Some remnant lamellar colonies were contained in the as-rolled and quenched microstructure due to a temperature drop and higher strain rate after hot-packed rolling. 

The current TiAl alloy contained more than 70 vol.% disordered α phase before each deformation, which played an important role in deformation. The rotation degree along the length of the lamellae was greater than that along the width, so α lamellae with a higher misorientation absorbed deformation, which further promoted the rotation and increased the misorientation along the length of α lamellae. Some α sub-grains formed along the length of α lamellae when the cumulative misorientation value reached a critical point.

Disordered α lamellae could not decompose into new α grains by themselves, but they could be decomposed by external factors. The grooved γ grains formed by the phase transformation from α lamellae during hot rolling cooperated with and decomposed α lamellae. When the γ grain diffused through α lamella, the parent α lamella was completely separated and ultimately transformed into non-continuous equiaxed α grains.

Deformation and phase transformation occurred during hot rolling. The α grains and γ grains around lamellar colonies underwent thorough recrystallization into fine recrystallized α grains and γ grains. These recrystallized α grains were transformed into γ grains and lamellar colonies during the rolling, accompanied by a temperature drop.

## Figures and Tables

**Figure 1 materials-14-04817-f001:**
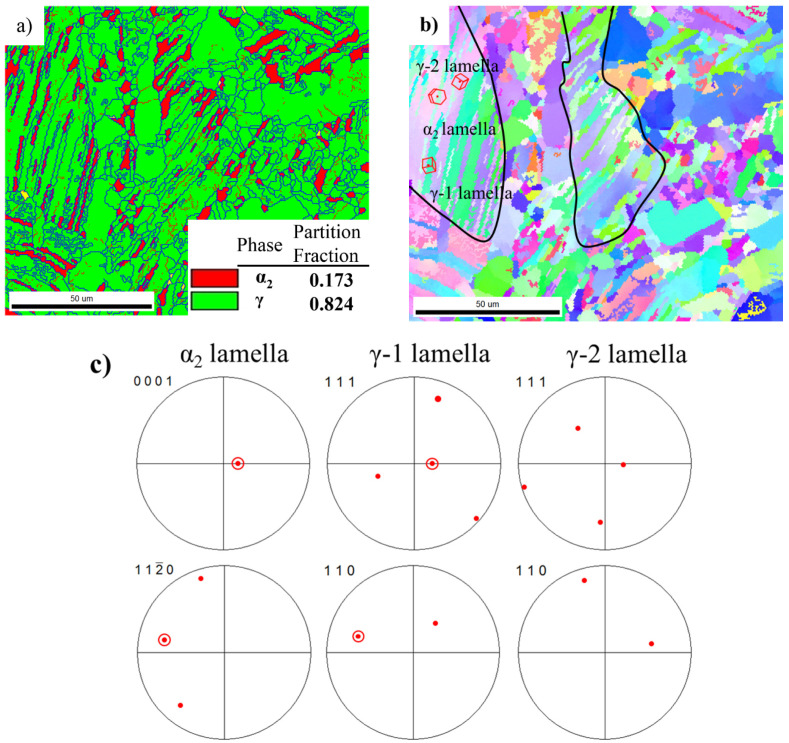
EBSD analysis of as-forged Ti-44Al-8Nb-(W, B, Y) alloy: (**a**) phase map, (**b**) orientation map and (**c**) pole figures of α_2_ lamella, γ-1 lamella, and γ-2 lamella.

**Figure 2 materials-14-04817-f002:**
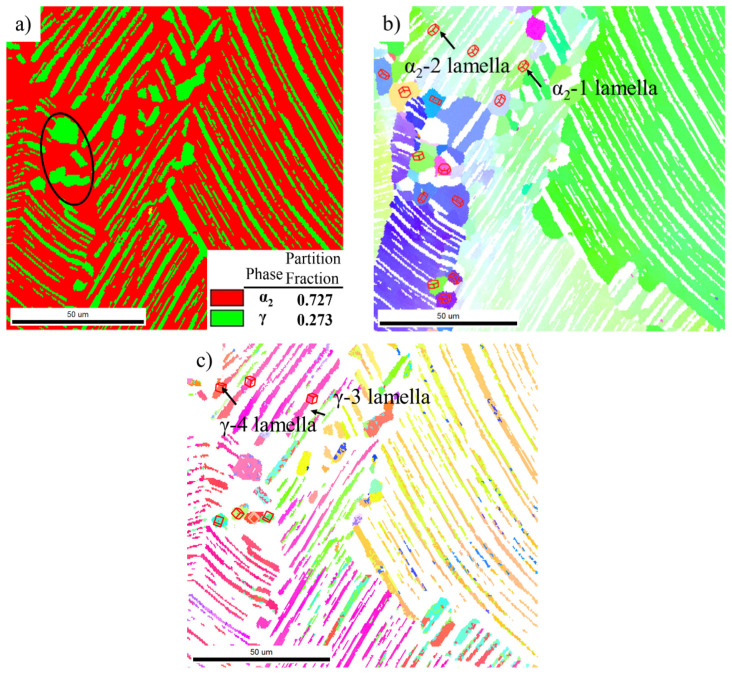
EBSD analysis of water quenched Ti-44Al-8Nb-(W, B, Y) alloy after annealing at 1250 °C for 2 h: (**a**) phase map, orientation maps of (**b**) α_2_ phase and (**c**) γ phase.

**Figure 3 materials-14-04817-f003:**
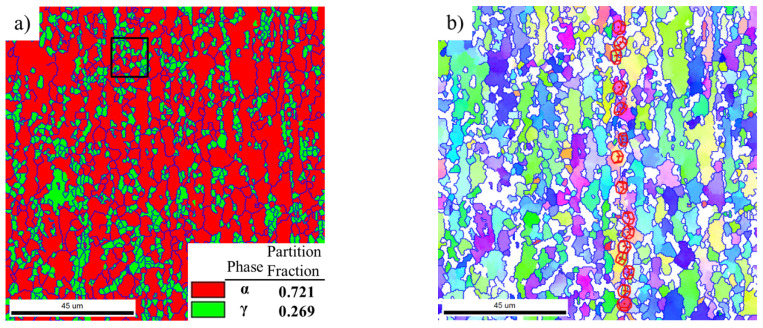
EBSD analysis of the quenched microstructure after three isothermal compression cycles at 1250 °C/0.1 s^−1^/25%: (**a**) phase fraction, orientation map of (**b**) α phase, (**c**) orientation map of the black region in (**a**,**d**) pole figure of each grain in (**c**).

**Figure 4 materials-14-04817-f004:**
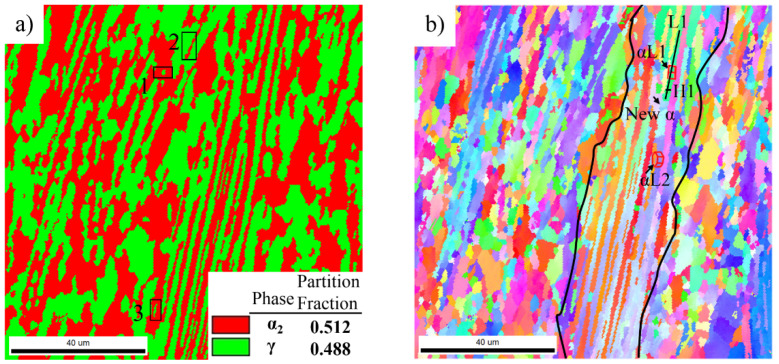
EBSD analysis of quenched rolled Ti-44Al-8Nb-(W, B, Y) alloy: (**a**) phase map and (**b**) orientation map.

**Figure 5 materials-14-04817-f005:**
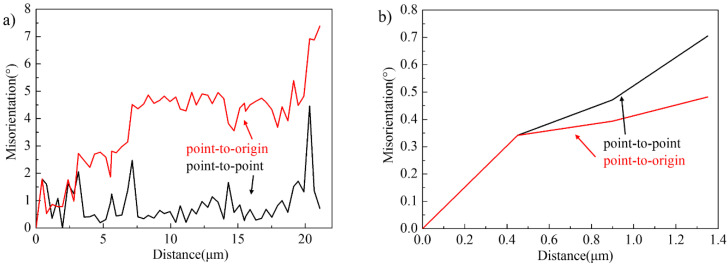
Misorientation measured along (**a**) L1 line and (**b**) H1 line in Figure 4b.

**Figure 6 materials-14-04817-f006:**
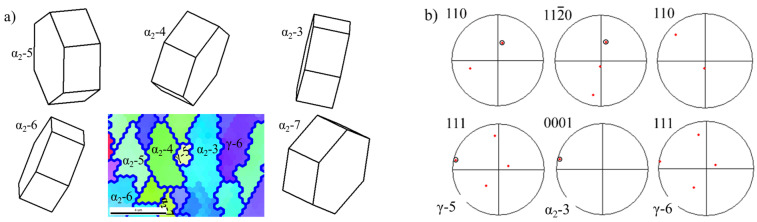
Local orientation maps and pole figures of the corresponding α_2_ and γ grains in the black rectangle in Figure 4b: magnification of (**a**), (**b**) region 1, (**c**) region 2, and (**d**) region 3.

**Figure 7 materials-14-04817-f007:**
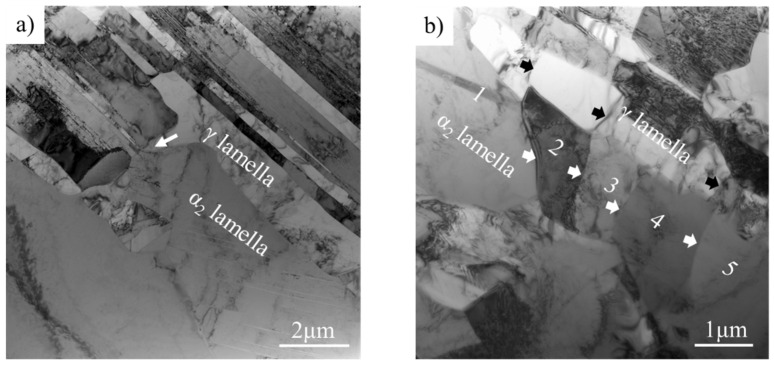
TEM analysis of α_2_ lamellae in water-quenched Ti-44Al-8Nb-(W, B, Y) alloy after three rolling passes: (**a**) α_2_ lamellae in a lamellar colony, (**b**) α_2_ sub-grains in α_2_ lamella, (**c**) grooving on the boundary of α_2_ lamella, (**d**) γ grain on the boundary of α_2_ lamellae, (**e**) equiaxed α_2_ grains, (**f**) the diffraction pattern of α_2_ grain, and (**g**) the final microstructure of the as-rolled alloy.

**Figure 8 materials-14-04817-f008:**
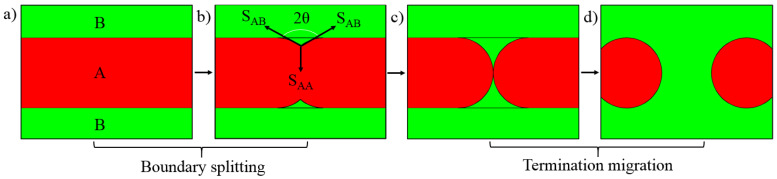
Schematic diagram of α lamellae decomposition: (**a**), (**b**) boundary splitting mechanism and (**c**), (**d**) termination migration mechanism.

**Figure 9 materials-14-04817-f009:**
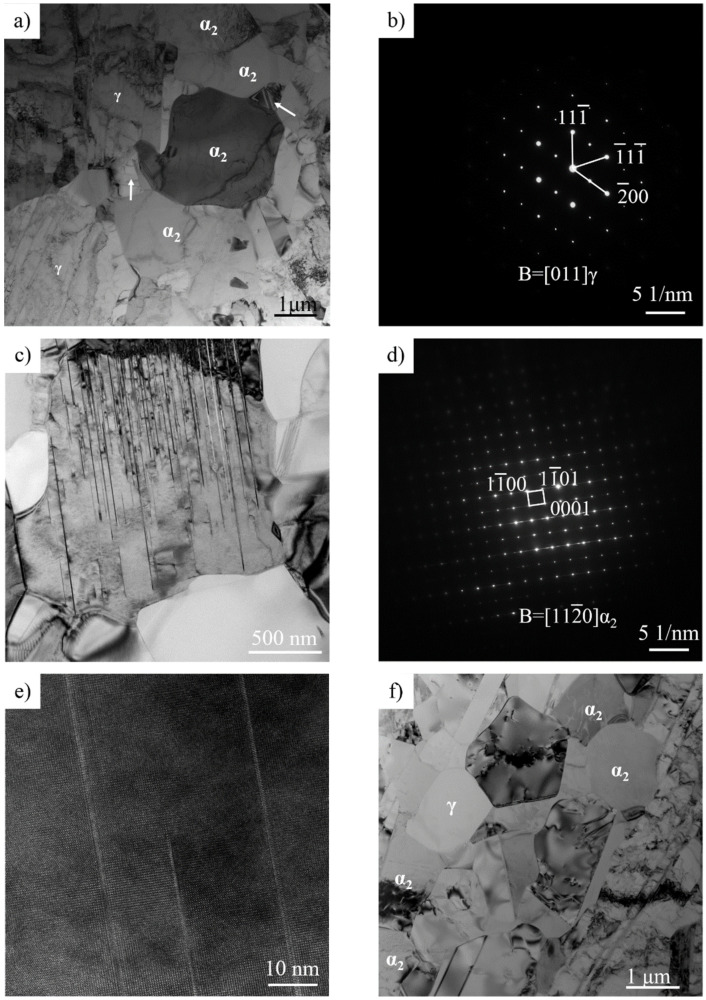
TEM analysis of the microstructure around the lamellar colony in as-quenched Ti-44Al-8Nb-(W, B, Y) alloy after three rolling passes. (**a**) The γ grains distributed at recrystallized α grains; (**b**) the diffraction pattern of γ grain; (**c**) the γ lamellae precipitated from α grains; (**d**) the diffraction pattern of α grain; (**e**) High resolution TEM images of the γ lamellae in α grain and (**f**) recrystallized α and γ grains around a lamellar colony.

**Figure 10 materials-14-04817-f010:**
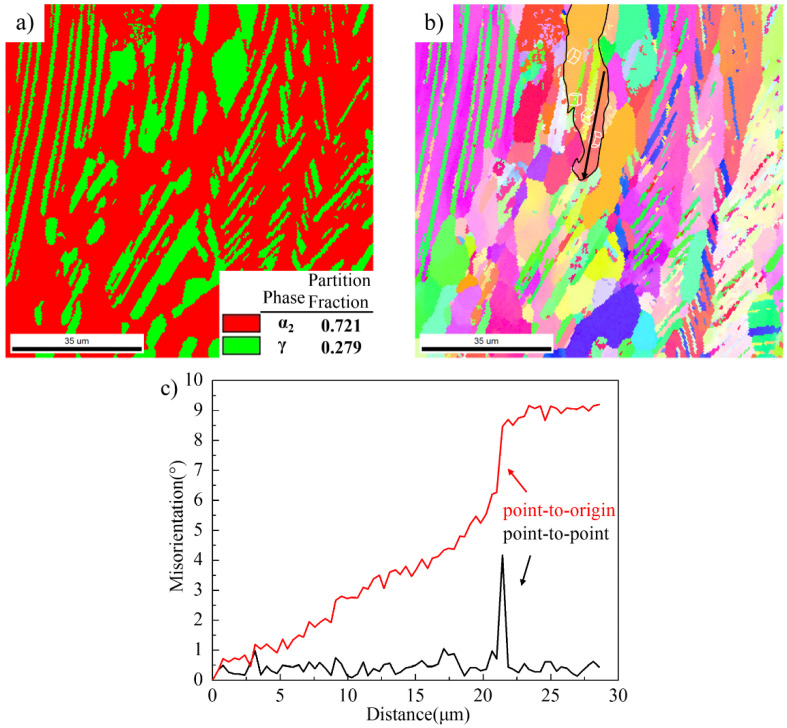
EBSD analysis of the microstructure of water quenched as-rolled Ti-44Al-8Nb-(W, B, Y) alloy with a total rolling reduction of 58% after annealing at 1250 °C for 15 min: (**a**) phase fraction, (**b**) orientation map, and (**c**) misorientation measured along the black line in (**b**).

**Figure 11 materials-14-04817-f011:**
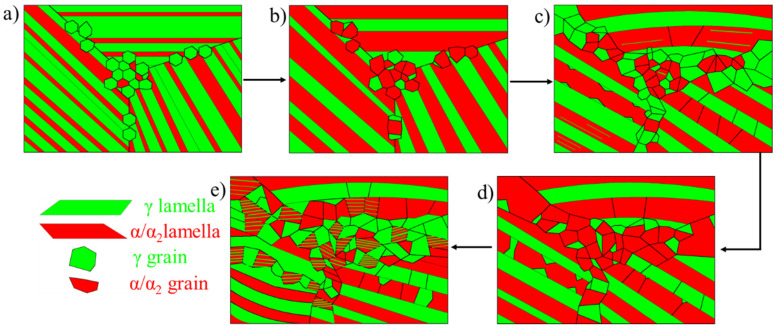
Microstructure evolution model of Ti-44Al-8Nb-(W, B, Y) alloy at 1250 °C during hot rolling. (**a**) Original microstructure, (**b**) microstructure after annealing by rolling for 2 h, (**c**) microstructure after the first rolling pass, (**d**) microstructure after annealing for 15 min between two rolling passes, and (**e**) microstructure after one more rolling pass.

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
