# Peer review of "Deformation and Phase Transformation of Disordered α Phase in the (α + γ) Two-Phase Region of a High-Nb TiAl Alloy"

_materials, 2021, doi:10.3390/ma14174817_

Round 1

Reviewer 1 Report

This manuscript deals with microstructure evolution of TiAl alloys during hot compression and hot packed rolling. The microstructural characteristics in each step was investigated by using EBSD and TEM analyses, and the mechanism behind evolution of the microstructures was revealed. As the authors mentioned in the manuscript, the deformation behavior of a phase is of great importance because it plays an significant role during deformation. 

1) English language and style must be revised. 

2) Overall, reviewers think the manuscript was not written in a friendly way It is not easy to follow the author's argument, and some figures does not directly explain the relevant sentences in the manuscript (e.x., orientation relationship, gamma2 lamella contains more LAGBs, how the authors distinguish newly formed a grain from other a grains and what are the newly formed a grains in Fig. 3c?).

3) Can the author discuss how the microstructure will change if the fraction of each phase in the initial state and processing temperature is changed?

4) Can the authors discuss the main findings of this work, and what can the readers learn from this work and apply them to future works ?

Reviewer 2 Report

This article deals with the description of deformation and phase transformations of as-forged TiAl alloy during the hot compression and hot packed rolling at temperatures characteristic for (α + γ) two phase region field.  Results are presented using appropriate methods, EBSD and TEM. Very solid work was done by a team of authors and the article also contains some new information regarding the development of the microstructure of TiAl alloy during thermomechanical processing. However, in my opinion, the article requires some speciifications and adjustments:

  • Nowhere in the article is the exact composition of the alloy given and only the marking Ti-44Al-8Nb- (W, B, Y) is used. In my opinion, the exact composition should be defined in the article, due to unclear content of B, W, Y whose content significantly affects the microstructure by the occurrence of borides, beta phase and yttrium-rich phases before forging. Especially borides could have a significant effect on recrystallization and without a known B content it is difficult to characterize phase transformations that could be affected locally.

  • The initial preparation of the alloy (melting or powder metallurgy) should also be at least briefly characterized and a more detailed description of the as-forged microstucture would also be appropriate. Did the microstructure contain beta phases, borides, or yttrium-rich phases? If so, were they determined using EBSD? These ambiguities should be characterized.

  • EBSD methodology is little described, the type of microscope used, step, type of electrolyte, polishing conditions are missing and also TEM should be more characterized (microscope, electrolyte, conditions).

  • The red markings in Figures 1 and 2 are not very visible, it would be appropriate to adjust it.

  • I also suggest inserting SE or BSE images of the analyzed microstructure into Figures 1-4, 10 before EBSD analysis. This could help readers gain greater insight and understanding.

Round 2

Reviewer 1 Report

All the issues were addressed in the revised version so that the manuscript could be acceptable.